

# Attribution of riming and aggregation processes by application of the vertical distribution of particle shape (VDPS) and spectral retrieval techniques to cloud radar observations

Audrey Teisseire[1], Anne-Claire Billault-Roux[3], Teresa Vogl[2], and Patric Seifert[1]

[1]Leibniz Institute for Tropospheric Research, Leipzig, Germany
[2]Leipzig Institute for Meteorologie (LIM), Leipzig University, Leipzig, Germany
[3]Environmental Remote Sensing Laboratory, Ecole Polytechnique Fédérale de Lausanne, Switzerland

**Correspondence:** teisseire@tropos.de

**Abstract.** Advancing the understanding of mixed-phase cloud microphysical growth processes requires a thorough detection of the transition processes from pristine hydrometeor states toward aggregates, rimed particles and graupel. In this study, a versatile combination of techniques is applied to detect and characterize aggregated and strongly rimed hydrometeors even under harsh atmospheric conditions such as the presence of orographic gravity waves. The approach combines dual-frequency observations from vertical-stare Doppler cloud radars as well as measurements from a polarimetric scanning cloud radar. Core of the approach are profiles of the Vertical Distribution of Particle Shape (VDPS) method that serve as a proxy for the presence of columnar, isometric, or prolate cloud particles. At height levels within the VDPS-based shape profiles where isometric particles are identified, Doppler spectra and dual-wavelength vertical-stare cloud radar observations are used to discriminate the occurrence of aggregation or graupel formation.

The underlying dataset was acquired in the framework of the 3-year field experiment, "Dynamic Aerosols Clouds and Precipitation Observation in the Pristine Environment in the Southern Ocean" (DACAPO-PESO) at the southern hemispheric midlatitude site of Punta Arenas, Chile (53°S, 71°W). The frequent presence of layers of supercooled liquid water and the permanent occurrence of orographic gravity waves motivate a strong interest to understand the formation of precipitation and the role of aggregation and riming at this site. Therefore, two case studies of both strong riming events and aggregation processes from the DACAPO-PESO campaign are presented to demonstrate the potential of combining the new VDPS retrieval with spectral methods, which analyze particle fall velocity and the coexistence of multiple particle types. We found that the identification of layers of supercooled liquid water is essential to pin down regions of riming in the observed cloud systems. In consequence, considering the general notion of the excess of liquid water in clouds over the southern hemisphere midlatitudes, our study serves as a preliminary investigation into the occurrence of riming and aggregation processes above Punta Arenas.

## 1 Introduction

Processes involving the growth of ice that lead to precipitating particles are crucial to understand, as over 60% of global precipitation reaching the surface is formed via the ice phase (Mülmenstädt et al., 2015; Heymsfield et al., 2020). The physics



of vapor-grown ice crystals, including plates, dendrites and columns, is well understood (Lamb and Scott, 1974; Libbrecht, 2003) in comparison to complex microphysical processes, including the formation of rimed particles, graupel, hail and aggre-
gates (Jiang et al., 2019). Temperature and supersaturation are involved in crystal growth and determine the shape or habits of hydrometeors. The overall structure of ice crystals grown in air can be classified into plate-like and columnar shapes as a function of temperature between $-40\,°C$ and $0\,°C$ (Bailey and Hallett, 2009). Indeed, in mixed-phase clouds existing at temperatures between $0\,°C$ and $-38\,°C$, the complex interaction of water vapor, ice, liquid droplets, vertical air motion and aerosol particles acting as cloud condensation nuclei (CCN) or ice nucleating particle (INP) (Pruppacher and Klett, 1996; Morrison
et al., 2012; Ansmann et al., 2019) results in a great variety of processes determining the distribution of the cloud thermodynamic phase. Understanding mixed-phase cloud processes such as aggregation and riming requires a thorough characterization of the liquid phase present in the cloud (Korolev et al., 2017). On the one hand, riming occurs in mixed-phase clouds when ice crystals are falling into a supercooled liquid cloud layer and collect droplets to form initially rimed particles, followed by graupel in the next stage (Locatelli and Hobbs, 1974; Field and Heymsfield, 2003). Rimed particles retain some of their
original shape while graupel particles are dense and almost spherical, falling faster than $1.5\,\mathrm{m\,s^{-1}}$ to $2\,\mathrm{m\,s^{-1}}$ (Kneifel et al., 2016; Kneifel and Moisseev, 2020; Vogl et al., 2022)). In the mid-latitudes, this represents an important precipitation formation process (Pruppacher and Klett, 1996; Lamb and Verlinde, 2011). On the other hand, aggregation occurs when ice particles are clumping together following collision as they fall to form snowflakes. This process produces aggregates, particles which are falling slowly compared to graupel because of their small density which results from the presence of cavities contained in the
aggregate structure (Houze, 1993).

There are certain needs for the availability of observational methods which aid to distinguish rimed from aggregated hydrometeors and which enable one to obtain information about the transition processes that occur prior to the formation of the graupel particles or aggregates. The reason is, that the partitioning of the ice and liquid phase in clouds varies regionally around the globe. One key region of interest is, e.g., the area of the Southern Ocean in the mid-latitudes of the southern hemisphere.
This region is known to feature an increased abundance of layers of supercooled liquid water, which is believed to be caused by a lack of a efficient ice nucleating particles (INP) over this region (Vergara-Temprado et al., 2018; Radenz et al., 2021). An excess of supercooled liquid water may lead to improved conditions for riming with the hypothesized consequence of a reduction in the abundance of aggregation events. An evaluation of observations for the occurrence of riming and aggregation in contrasting environments can potentially be utilized to challenge the representation of riming and aggregation processes
in general circulation models, as these two hydrometeor classes are amongst the most important ones in many model setups (Seifert and Beheng, 2006).

The increasing availability of data from advanced multi-frequency Doppler radar systems has facilitated the development of algorithms to discriminate rimed particles from aggregates utilizing various techniques, including dual-frequency reflectivity (Matrosov, 1998), triple-frequency reflectivity (Leinonen et al., 2018; Tridon et al., 2019; Kneifel et al., 2015; Mroz et al.,
2021; von Terzi et al., 2022), dual-frequency reflectivity combined with Doppler measurements (Mason et al., 2018), and comprehensive Doppler spectral information (Mróz et al., 2021). For example, Kneifel et al. (2015) and von Terzi et al. (2022) suggest to apply a radar triple frequency method (W-, $K_a$- and X-band radars) to differentiate aggregates from rimed particles



and graupel. Another way to discriminate different hydrometeor populations is the separation of peaks in cloud radar Doppler
spectra using observations of vertically pointing cloud radar systems (Kalesse et al., 2019; Radenz et al., 2019; Vogl et al.,
2024). However, this technique is limited, e.g., with respect to atmospheric turbulence, which broadens the spectra and makes
the detection and separation of peaks difficult or even impossible. A method was developed using Artificial Neural Networks
(ANN) to predict riming from Doppler cloud radar observations (Vogl et al., 2022). This approach is based on the radar
reflectivity factor, the width from left to right edge of the spectrum above the noise floor and the skewness as input features. This
method is limited to unambiguous spectral riming signatures and thus to cases of strong riming. A recent approach aims to use
the Slanted Linear Depolarization Ratio (SLDR) as unique polarimetric parameter using a SLDR-mode scanning cloud radar
to derive the vertical distribution of particle shape (VDPS method, Teisseire et al. (2024)) based on earlier studies (Myagkov
et al., 2016a; Matrosov et al., 2012). This method, which focuses solely on deriving particle shape, lacks the capability to
distinguish between graupel and aggregates, as both of which are derived as isometric particles.

In this study, we propose to complement the VDPS method with other techniques using spectrograms and lidar measure-
ments to differentiate strong riming and aggregation processes in the complex atmospheric environment above the southern
hemispheric midlatitude site of Punta Arenas, Chile. The presented approach assumes that riming processes require super-
cooled liquid droplets to occur, in contrast to aggregation processes, which can occur in a relatively dry environment but are
not limited to such conditions. We acknowledge that this technique can solely be used to distinguish aggregation events from
strong riming events, where graupel particles are formed. The detection of earlier stages of riming is not in the scope of this
study because slightly rimed particles initially retain their original shape and therefore do not exhibit the characteristic sig-
nature of isometric particles in the SLDR scans. Indeed, the VDPS method, described in Section 3.1, is able to derive the
apparent particle shape as a function of height while spectrograms and lidar measurements are helpful to detect supercooled
liquid droplets. Dual wavelength ratio (DWR) between $K_a$- and W-band radars will be calculated as a proxy of particle size.
The paper is structured as follows. Section 2 gives an overview about the field campaign and instrumentation, Section 3 de-
scribes the VDPS method and Section 4 presents four case studies presenting strong riming and aggregation processes. Finally,
Section 5 discusses limitations and possible extensions of the new approach and concludes the study.

## 2 Datasets

### 2.1 DACAPO-PESO campaign

The DACAPO-PESO (Dynamics, Aerosol, Cloud And Precipitation Observations in the Pristine Environment of the Southern
Ocean) field campaign was conducted to obtain a dataset of long-term ground-based supersite remote sensing observations
of aerosols and clouds for the midlatitude site of Punta Arenas, Chile (Radenz et al., 2021). Punta Arenas is located in the
southern-hemispheric midlatitudes at the southernmost tip of the South-American continental land mass. DACAPO-PESO was
a joint initiative of Leibniz Institute for Tropospheric Research (TROPOS), University of Magallanes (Punta Arenas, Chile),
and University of Leipzig (LIM). It was dedicated to exploring the dynamics and interactions between aerosols and clouds in
the atmosphere. In terms of aerosol conditions, pristine marine air masses are prevalent, primarily because of the consistent





westerly winds (Schneider et al., 2003; Foth et al., 2019; Jimenez et al., 2020; Floutsi et al., 2021; Radenz et al., 2021) at a latitude where no other landmasses exist in the southern hemisphere. Punta Arenas is occasionally influenced by orographically driven gravity waves due to the strong westerly winds moving above the Andes Cordillera which need to be accounted for in cloud studies (Alexander et al., 2017; Silber et al., 2020; Radenz et al., 2021; Vogl et al., 2022). Finally, stratospheric smoke
events have been studied using lidar measurements during the DACAPO-PESO campaign (Ohneiser et al., 2020).

The Leipzig Aerosol and Cloud Remote Observations System (LACROS, Radenz et al. (2021)),was deployed and operated for 3 years from 27 November 2018 until 27 November 2021. During a subset of the campaign from November 2018 through September 2019, LIMRAD94, a 94-GHz cloud radar, was deployed next to LACROS by the Leipzig Institute for Meteorology (LIM) of the University of Leipzig. The site and the DACAPO-PESO experiment have been described in more detail in Floutsi
et al. (2021) and Radenz et al. (2021).

## 2.2   Instruments

The primary set of instruments that was utilized in the framework of this study is listed in Table 1. LACROS encompasses a suite of remote sensing instrumentation including a 35-GHz SLDR-mode scanning cloud radar, and a lidar of type PollyXT, which are used in this study. Measurements from a modified version of the 35-GHz cloud radar MIRA-35, which is operated
in SLDR-mode (Reinking et al., 2002; Matrosov et al., 2012) are the prerequisite for the VDPS method deployed in the scope of this study (Teisseire et al., 2024). In contrast to the standard LDR mode, variations in the orientation of hydrometeors only have small effects on the measured SLDR, even at low elevation angles (Matrosov, 1991; Matrosov et al., 2001). MIRA-35 is a dual-polarization radar which emits linearly polarized pulses of radiation through the co-channel, while the returned signals are received in both the co- and cross-channels. The properties of the standard LDR-mode MIRA-35 are elaborated in detail
in Görsdorf et al. (2015). The SLDR-mode radar is implemented based on a traditional LDR-mode radar with a $45°$ rotation of the antenna emission plane. This rotation allows polarimetric measurements to be more sensitive to the particle shape as the oscillation of particles during falling becomes negligible (Matrosov, 1991; Bringi and Chandrasekar, 2001).

In the framework of the presented study, MIRA-35 was steered towards geographic south direction and performed RHI scans from $90°$ (zenith pointing) to $150°$, corresponding to $30°$ elevation over the horizon towards north direction, at an
angular velocity of $0.5°\text{s}^{-1}$ for obtaining polarimetric signatures of the hydrometeors present in the observation volume. This notation of the elevation angle range will be used throughout this article.

LIMRAD94 is a zenith-pointing Frequency-Modulated Continuous Wave Dual-Polarization radar manufactured by Radiometer Physics GmbH (RPG-FMCW-DP) 94-GHz radar and operates at 3.2-millimeter wavelength. The radar utilizes frequency modulated continuous wave (FMCW) signals and provides vertical profiles of spectrally resolved radar variables which
can be used to identify atmospheric scatterers such as cloud particles, raindrops, snowflakes and insects. On the one hand, LIMRAD94 measurements are utilized for calculation of the Dual-Wavelength Ratio (DWR) and for generating Doppler spectrograms of spectral reflectivity ($sZ_e$, unit: $1\,\text{dBsZ} = 10\log_{10}(1\,\text{mm}^6\,\text{m}^{-3}\,(\text{m}\,\text{s}^{-1})^{-1}))$. On the other hand, this radar is used to deliver the vertical-stare $Z_e$ and Mean Doppler Velocity (MDV) during periods of Range Height Indicator (RHI) scans of MIRA-35 (see Tab. 1).



Finally, a 5° off-zenith pointing lidar of type PollyXT completes the instrumentation setup (Engelmann et al., 2016). PollyXT provided for this study the cross-polarized component of the attenuated backscatter at a 532-nm wavelength, from which the volume depolarization ratio (VDR) is calculated, a parameter used in the detection of supercooled liquid layers (Table 1). During DACAPO-PESO, PollyXT also delivered valuable VDR and backscatter observations for the long-term mixed-phase cloud statistics that were reported by Radenz et al. (2021). The full multi-wavelength capability of PollyXT was not required

for the current study, but it was used by Floutsi et al. (2021) for the characterization of aerosol layers which were observed above Punta Arenas during DACAPO-PESO.

General information about the uncertainties in the parameters retrieved by the various instruments and methods are given in the references that are provided in Table 1. Only a few critical data quality aspects will be outlined in the following. For the VDPS retrieval (see Sec. 3.1) the purity of the SLDR and the value of the Signal-to-Noise Ratio (SNR) are of high relevance.

The better the decoupling between the co- and cross-channels and the deteceted SNR of the MIRA-35 receiver is, the more accurate is the retrieval of the different particle habits. For Mira-35 in SLDR mode, Teisseire et al. (2024) gave a co-cross-channel isolation of $-35$ dB. Also for LIMRAD94, the polarimetric decoupling has to be considered. It was found to be of less quality than the one from Mira-35. Therefore, it was decided to use the SLDR values from Mira-35 in the remainder of this study.

Our study does not rely on a good absolute calibration of $Ze$ of both cloud radars. Only the relative calibration was required to be constant over time, which was the case according to regular tests. Methods for Doppler spectral analysis require a good zenith-alignment of the antennas to provide accurate Doppler spectra of the vertical motion of the observed hydrometeors. Vertical alignment was perfectly assured for LIMRAD94 based on an internal levelling sensor. For MIRA-35, which is attached to the superstructure of a measurement container, alignment was not perfect. In the remainder of this study, this issue is

considered in the interpretation of Doppler spectra from MIRA-35.

From the PollyXT lidar, only qualitative information about the volume depolarization ratio (VDR) was required for this study in order to identify layers of liquid water by the occurrence of relatively low VDR. Specific uncertainties, as they are required for detailed aerosol studies (Engelmann et al., 2016), have not been taken into account.

## 3   Methods

In this study we will combine the VDPS method (Section 3.1) with spectral approaches. The VDPS method is able to deliver the vertical distribution of particle shape in a cloud using the polarimetric parameter SLDR and gives hints on the types of microphysical processes occurring in the observed cloud. By combining the VDPS method with spectral techniques and lidar measurements, the overall methodology can be reinforced to enable a solid approach for the discrimination between graupel and aggregates. In this Section, the VDPS method and the $\text{DWR}_{\text{Ka}-\text{W}}$ calculation are briefly outlined. A third part is dedicated

to present an introduction to riming and aggregation processes.



**Table 1.** Technical characteristics of the instruments described in Sec. 2.2 during the deployment of DACAPO-PESO in Punta Arenas, Chile. The table lists only the parameters utilized in this study, although the instruments are able of providing additional parameters.

| Data source (Reference) | Frequency $\nu$ Wavelength $\lambda$ | Measured / retrieved quantity | Temporal resolution | Vertical range | Vertical resolution |
|---|---|---|---|---|---|
| **SLDR-mode scanning cloud radar MIRA-35** Metek company MIRA-35-SLDR (Görsdorf et al., 2015; Teisseire et al., 2024) | $\nu = 35.2\,\text{GHz}$ | Signal-to-noise ratio SNR Slanted linear depolarization ratio SLDR Radar reflectivity factor $Z_e$ | 1 s | 150 – 15000 | 31.18 m |
| **Doppler cloud radar LIMRAD94** RPG-FMCW-94-DP (Küchler et al., 2017) | $\nu = 94\,\text{GHz}$ | Spectral reflectivity $sZ_e$ Radar reflectivity factor $Z_e$ Mean Doppler velocity $\bar{v}_D$ Spectrum width $\sigma_w$ Linear depolarization ratio LDR | 5 s | 120 – 12000 m | 30 – 45 m |
| **Multiwavelength Raman polarization lidar PollyXT** (Engelmann et al., 2016) | $\lambda = 532\,\text{nm}$ | Volume depolarization ratio VLDR | 30 s | 100 – 40000 m | 7.5 m |
| **Weather model forecast** ECMWF IFS (Owens and Hewson, 2018) | | Temperature $T$ Pressure $P$ Relative Humidity RH | 3600 s | 10 – 12000 m | 20 – 300 m |

## 3.1 Vertical Distribution of Particle Shape (VDPS)

The VDPS method aims to characterize the shape of cloud particles from SLDR-mode scanning cloud radar observations. This approach combines values from a scattering model developed by Myagkov et al. (2016a) and measurements of SLDR at different elevation angles $\theta$. The spheroidal scattering model delivers the polarizability ratio ($\xi$) and the degree of orientation

($\kappa$), which describe the apparent particle shape by means of a density-weighted axis ratio and their preferred orientation, respectively. The VDPS method uses only the polarizability ratio $\xi$ because of its dependency on the apparent particle shape. Then, it sorts particles into three primary categories based on their shape (oblate, isometric or prolate particles). The oblate and prolate spheroid shapes are described in more detail in Myagkov et al. (2016b). A polarizability ratio $\xi \approx 1$ (when the polarizability ratio $\xi$ takes values between $0.8$ to $1.2$) corresponds to isometric particles characterizing spherical particles or

particles with low density. In this study, we consider particles as isometric when they do not produce significant polarimetric signatures. Such particles have either spherical or just slightly-non-spherical shapes. In addition, non-spherical particles with low density (low-refractive index) also appear to be isometric (Myagkov et al., 2016b; Teisseire et al., 2024). On the contrary, a polarizability ratio $\xi < 0.8$ and $\xi > 1.2$ describe oblate and prolate particles, respectively. This method is unique in that it utilizes only SLDR at minimum (close to $90°$, denoted as $\theta_{\min}$) and maximum (close to $150°$, denoted as $\theta_{\max}$) elevation





angles to calculate the polarizability ratio $\xi$. Because the VDPS method relies on polarimetric measurements at different elevation angles, horizontal homogeneity of the observed clouds is required, which is tested based on visual inspections of the spatial homogeneity of the utilized RHI scans of the input variables SNR and SLDR. The approach works in a three-step procedure. (i) The RHI scans of SLDR are filtered for noise artifacts and the linear regression of SLDR vs. $\theta$ is obtained with a height resolution of 30 m. (ii) Simulations of SLDR vs. $\theta$ are used to identify the possible particle shape classes, and associated

uncertainties are assessed. (iii) The linear regression $\frac{\partial \text{SLDR}}{\partial \theta}$ calculated in (i) is deployed to assign the polarizability ratio $\xi$ and quantify the primary particle shape class. Finally, the VDPS method provides the vertical profile of the polarizability ratio $\xi$, describing the apparent particle shape, for all heights where cloud layers, fulfilling the criteria of homogeneity and signal quality, are observed. The vertical distribution of particle shape (i.e., polarizability ratio $\xi$) is valuable as it provides insights into processes such as transformation, stratification, and hydrometeor sedimentation from the upper to lower regions

of a cloud. This approach was implemented by means of an automatized framework and for a long period of measurements, covering several field campaigns (DACAPO-PESO in Punta Arenas, Chile and CyCARE in Limassol, Cyprus). The VDPS method is detailed in Teisseire et al. (2024), where the approach is validated by means of three case studies representing the three primary particle shape classes, prolate, isometric and oblate particle shapes.

## 3.2 Dual wavelength ratio (DWR) calculation

Here, we perform the following steps to compute $\text{DWR}_{\text{Ka}-\text{W}}$. Firstly, the observed reflectivity $Z_e$ from MIRA-35 and LIM-RAD94 are brought to the same time-height grid using nearest-neighbor interpolation. The attenuation by atmospheric gases, including water vapor, is estimated using the Passive and Active Microwave TRAnsfer (PAMTRA) tool (Mech et al., 2020) and the profiles of the temperature and humidity from European Centre for Medium-Range Weather Forecasts (ECMWF) Integrated Forecast System (IFS). Finally, MIRA-35 reflectivity is offset-corrected to match LIMRAD94 reflectivity in the

Rayleigh regime following the method proposed by Dias Neto et al. (2019). For 30-minute time intervals, the parts of the cloud at least 1000 m above the 0 °C isotherm, and 4000 m above the surface, where $Z_{e(\text{Ka})}$ from MIRA-35 is ranged between $-30$ and $-10$ dBZ, are extracted. In these areas, both cloud radar instruments are expected to be in the Rayleigh regime. The difference in the observed reflectivity $Z_{e(\text{W})} - Z_{e(\text{Ka})}$ is thus the offset, which is added to $Z_{e(\text{Ka})}$ for the respective 30-minute period. The quality of the offset correction is ensured by the requirement that the correlation of $Z_{e(\text{Ka})}$ and $Z_{e(\text{W})}$ is well above

0.9 for the selected cloudy pixels.

## 3.3 Riming and aggregation detection

Riming and aggregation are distinct processes, yet it can be challenging to distinguish them in radar measurements. Their microphysical properties are summarized in Table 2. Riming processes take place when ice crystals fall into a supercooled liquid cloud layer, initially producing rimed particles, which later evolve into graupel, i.e., spherical and dense particles. In this

study, we will focus on graupel formation through the riming process, as graupel are isometric particles. Aggregation occurs when ice crystals collide and adhere to each other, forming aggregates, particles with low density. The VDPS method alone is not able to differentiate these two processes with precision because both of them form at advanced stages isometric particles



(spherical particles in the case of graupel and particles which do not produce considerable polarimetric signatures because of their low density as in the case of aggregates). In this study we assume that riming processes require supercooled liquid

droplets to produce graupel. If a supercooled liquid layer can be identified from low depolarization in lidar observations, or if a secondary slow-falling mode is visible in the Doppler spectrogram, this is a strong indication that isometric particles detected below result from a riming event. On the contrary, isometric particles associated with a liquid-subsaturated environment will have originated more likely from an aggregation event, producing particles with low density, which are therefore considered as isometric. In addition, the DWR serves as an indicator of particle size. DWR is an effective parameter to demonstrate the

increase in particle size and to characterize large aggregates (Matrosov et al., 1992; Matrosov, 1998; Matrosov et al., 2022; von Terzi et al., 2022). Furthermore, the fall velocity of particles is often used in the literature to describe riming processes. If ice particles are falling faster than $-2 \, \mathrm{m \, s^{-1}}$, the process is usually attributed to riming. However, the Doppler velocity is not easily interpretable in certain cases such as in the presence of strong orographic gravity waves, as it was occasionally the case in Punta Arenas during the DACAPO-PESO campaign (see Section 2.1). Finally, riming and aggregation are strongly

dependent on the temperature range (Kneifel and Moisseev, 2020). Waitz et al. (2022) showed that riming is most prevalent at temperatures around $-7\,°C$. Additionally, Hallett-Mossop rime splintering describes a process where liquid droplets collide with large ice crystals, freeze and shatter in a temperature range between $-3\,°C$ and $-8\,°C$ producing graupel and a plethora of small ice splinters (Hallett and Mossop, 1974; Atlas et al., 2022; Korolev et al., 2022). Concerning aggregation processes, experimental studies have identified that strongest aggregation occurs in the range from $-10\,°C$ to $-15\,°C$ (Hobbs et al., 1974).

Therefore, the use of the temperature in this article is also helpful to support the differentiation between riming and aggregation processes.

**Table 2.** Microphysical properties of riming and aggregation processes in relation to detection techniques, summarizing Sec. 3.3.

| Process | Advanced stage of riming | Aggregation |
|---|---|---|
| Particles | Graupel | Aggregates |
| Shape / density (VDPS) | Spherical / High density (isometric) | Diverse shapes possible / Low density (isometric) |
| Size (DWR) | large particles | very large particles |
| Fall velocity (MDV, Doppler spectrograms) | $< -2 \, \mathrm{m \, s^{-1}}$ (Kneifel and Moisseev, 2020) | $\approx -1 \, \mathrm{m \, s^{-1}}$ |
| Supercooled liquid droplets (lidar, Doppler spectrograms) | Yes | Not necessary |
| Temperature (ECMWF-IFS) | Favored at warmer temperature ($> -10\,°C$) (Kneifel and Moisseev, 2020) | Favored at colder temperature ($\approx -15\,°C$) (Connolly et al., 2012) |





## 4 Results

This section aims to show the applicability of the VDPS method combined with spectral techniques to differentiate strong riming and aggregation processes by means of four case studies. In this study, we will focus solely on the advanced stage of the riming process, which produces graupel (see Sec. 3.3). Particles in an earlier riming state are in fact neither spherical nor low-density particles and would not be detected as isometric particles by the VDPS method, as was pointed out above. As mentioned in Section 3.3, these two processes produce isometric particles, which makes their differentiation challenging based on the VDPS method only. The VDPS retrieval delivers the vertical distribution of the polarizability ratio $\xi$ while the Doppler spectrograms recorded by LIMRAD94 ($sZ_e$) and MIRA-35 (SLDR) (see Section 2.2) will provide information about the fall velocity of particles, the size of particles and the presence of supercooled liquid droplets or secondary ice production (SIP) during the studied events. The presence of supercooled liquid cloud layers is confirmed using the volume depolarization ratio (VDR) measured by the lidar (Table 1). In all spectrograms, we observe a shift in Doppler velocity of approximately $0.5\,\mathrm{m\,s^{-1}}$ when comparing LIMRAD94 and MIRA-35 measurements. This effect is attributed to the misalignment of MIRA-35. For the purposes of this study, only the Doppler velocity values from LIMRAD94 will be considered. The MDV is used in this study to quantify the impact of the gravity waves during the DACAPO-PESO campaign. The vertical distribution of $Z_e$ from cloud top to cloud base is derived from LIMRAD94 measurements and computed concurrently with the spectrograms over the same time range. Finally, the $DWR_{Ka-W}$ will aid to determine the evolution of the size of particles (von Terzi et al., 2022). Four case studies were selected from the DACAPO-PESO campaign, introduced in Section 2.1, and for the time period during which MIRA-35 and LIMRAD94 were collocated. All cases were checked for homogeneity from $90°$ to $150°$ elevation angle. In this section, we will describe clouds from top to bottom to be coherent with the pathway of riming and aggregation processes.

### 4.1 Riming detection

The following section includes two case studies where riming was observed during DACAPO-PESO.

#### 4.1.1 Riming case study on 23 August 2019, 14:30 UTC

Figure 1 shows a deep stratiform precipitating cloud from 13:00 to 16:00 UTC. In Fig 1a, a lowering of the melting layer can be observed between 13:50 and 15:20 UTC, recognizable by an increase of $Z_e$ in the rain at around $0.8$ km height, suggesting that particles require additional time to undergo melting. Dense particles such as graupel have the potential to contribute to the descent of the melting layer influencing the bright band (Li et al., 2020). Figure 1c demonstrates that stable patterns of negative MDV occurred within the cloud. This suggests that no gravity wave activity is evident, allowing to use values of the MDV for microphysical interpretations in this case study. The RHI scan of SLDR from $90°$ to $150°$ elevation angle is presented in Fig. 2a and is corresponding to the second white band at around 14:30 UTC in Fig. 1b. Finally, spectral signatures of the event are illustrated in Fig. 3 where Figs. 3a and 3c are based on observations at the beginning of the RHI scan while Figs. 3d and 3f were recorded at the end of the RHI scan.




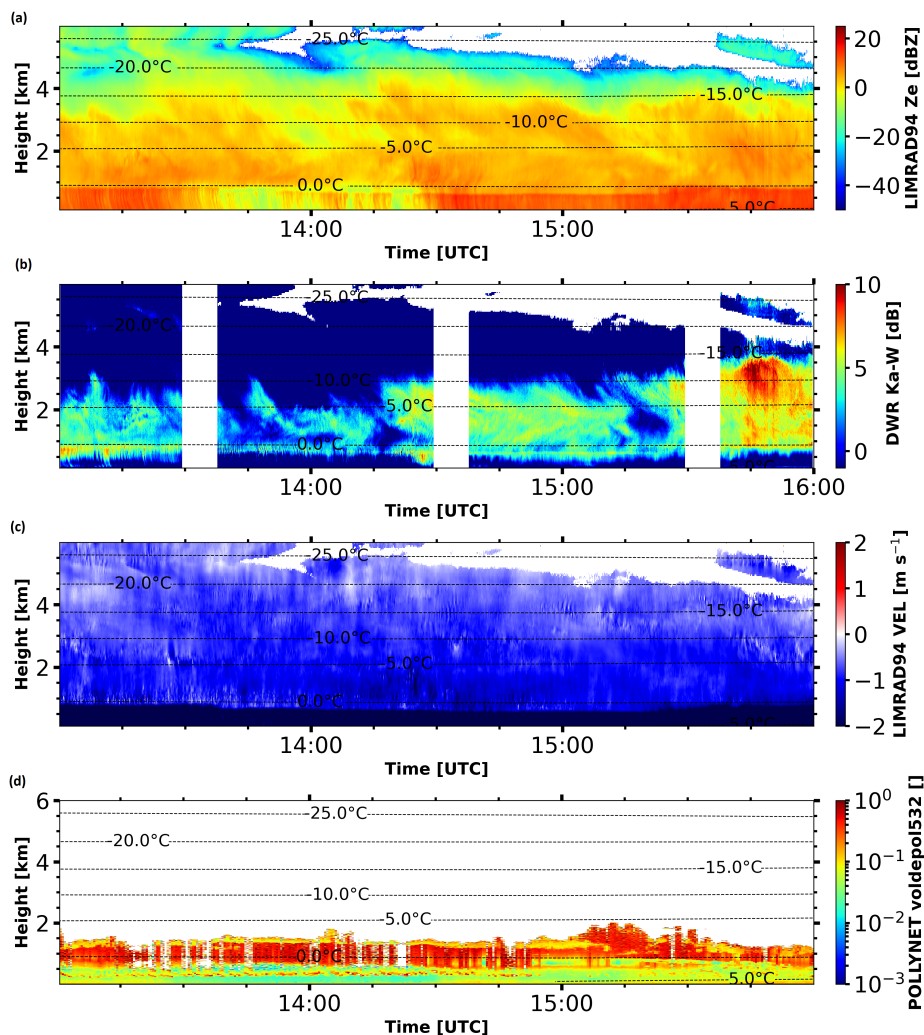

**Figure 1.** Case study of a deep mixed-phase cloud event observed with multi-wavelength polarimetric cloud radars at Punta Arenas, Chile, on 23 August 2019 from 13:00 UTC to 16:00 UTC. (a) Vertically pointing W-band (94 GHz) radar reflectivity factor $Z_e$, (b) represents the dual-wavelength ratio (DWR$_{Ka-W}$, ratio of $Z_e$ between Ka-band MIRA-35 and W-band LIMRAD94), and (c) Mean Doppler velocity. No measurements are recorded in the time periods where white bands are shown in (b), due to RHI and PPI scans, that were conducted during these periods. (d) The volume depolarization ratio (VDR) is measured with the lidar to detect the possible supercooled liquid droplets. In (a), (b), (c) and (d) isolines of ECMWF-IFS air temperature are plotted by means of dashed lines.



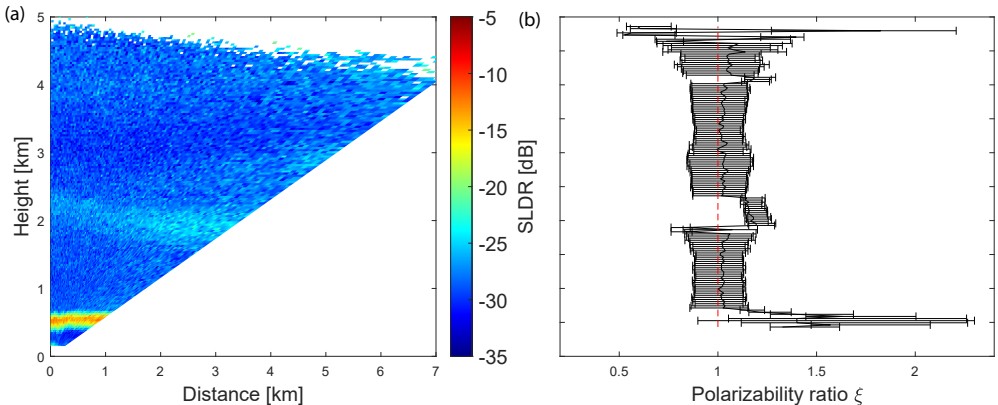

**Figure 2.** (a) RHI-scans of SLDR from $90°$ to $150°$ elevation angle, observed on 23 August 2019, recorded from 14:29:05 to 14:31:33 UTC, in Punta Arenas. (b) Vertical distribution of the polarizability ratio $\xi$ derived from the VDPS method and correlated with the RHI scan presented in (a)

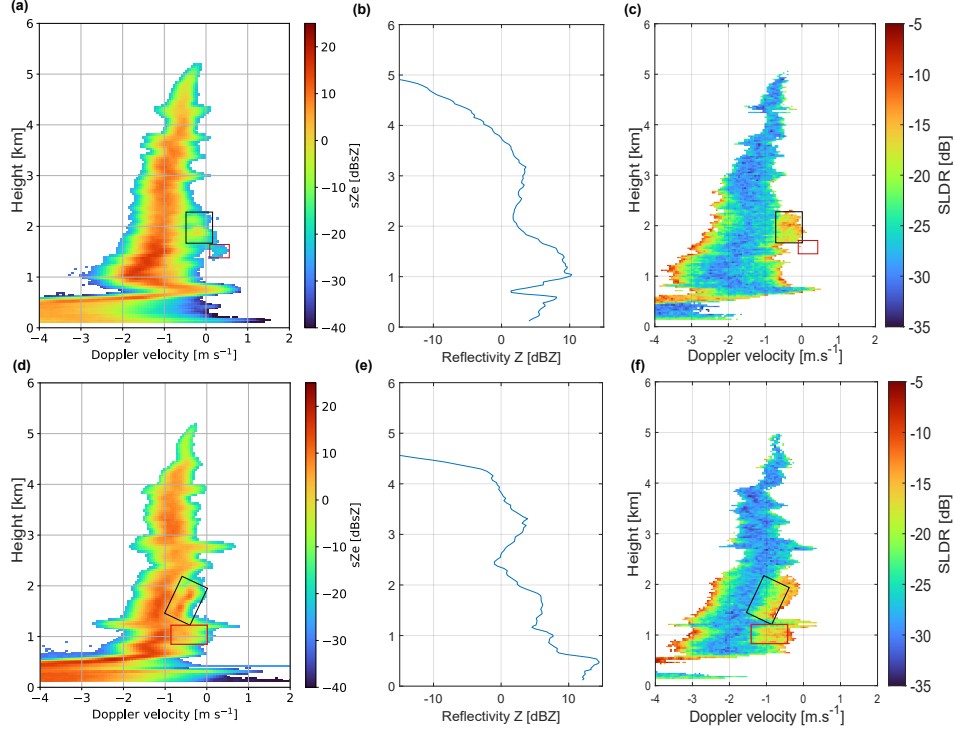

**Figure 3.** Spectrograms of (a), (d) $sZ_e$ recorded by LIMRAD94 at 14:29:16 and 14:33:54 UTC, respectively, and (c), (f) SLDR recorded by MIRA-35 at 14:30:10 and 14:32:56 UTC, respectively, on 23 August 2019 in Punta Arenas, Chile. (b) and (e) represent the profiles of $Z_e$ at 14:29:16 and 14:33:54 UTC, respectively.



In Fig. 2a, the melting layer is clearly visible at around 0.7 km height, recognizable by high values of SLDR (red layer). Above the melting layer, elevated values of SLDR of approximately $-23$ dB are observed over all elevation angles from 2.5 to 2 km height. Within this layer, the VDPS retrieval, shown in Figure 2b, yields polarizability ratio $\xi > 1$, corresponding to prolate particles. The temperature range in this region is between $-10\,°C$ and $-5\,°C$ (see Fig. 1), which is favoring the growth of columnar crystals (Libbrecht, 2017). Between this specific layer associated to prolate particles at approximately 2 km height and the melting layer at around 0.7 km height, particles exhibit minimal depolarization, as evidenced by notably low SLDR values (Fig. 2a), characterized by a polarizability ratio $\xi \approx 1$ (Figure 2b). Correspondingly isometric particles are present in this height range. The abrupt transition from prolate to isometric particles is observable at around 2 km height. In Figure 3a and 3c, a secondary slow-falling mode associated with the rise in $Z_e$ of the main hydrometeor population is starting at an altitude of 2 km height: a secondary spectral mode (indicated with a black box) is initially identified between 2.2 km and 1.8 km height, characterized by a fall velocity near $0$ m s$^{-1}$ and a s$Z_e$ of around 0 dBsZ. This mode is well represented in Figure 3c by high values of SLDR around $-15$ dB, indicating the presence of another population of columnar crystals, depolarizing strongly at zenith-pointing. Another distinct secondary spectral mode (enframed with a red box) is noticeable between 1.8 km and 1.5 km height, where particles have a Doppler velocity exceeding $0$ m s$^{-1}$ and lower s$Z_e$ of approximately $-20$ dBsZ. This third spectral mode is not correlated with any measurable value of SLDR, which gives hint that supercooled liquid water droplets were present. Indeed, the signal received in the cross-polar channel of the supercooled liquid water contained in a cloud is usually too weak to be detected. This conclusion is corroborated by the presence low values of lidar VDR in Fig. 1d at around 1.6 km height, characterized by a yellow layer and followed by an extinction of the signal just above this layer. This low depolarization layer is a result of backscattering by supercooled liquid droplets. A second set of spectrograms of s$Z_e$ and SLDR measured just after the end of the RHI scan of MIRA-35 (14:30:10 and 14:32:56 UTC) is shown in Fig. 3d and 3f. In both panels, a secondary spectral mode associated to high values of s$Z_e$ and SLDR, of approximately 10 dBZ and $-13$ dB, respectively, is shown between 2.1 km and 1.3 km height, as indicated by the black rectangle. Such elevated values of s$Z_e$ and corresponding high values of SLDR recorded at zenith pointing are representative for the presence of another population of ice crystals such as columnar crystals which are falling slowly compared to the main hydrometeor population. The formation of the secondary population of columnar crystals coincides with an increase in $Z_e$ for the primary hydrometeor population (Figs. 3b and 3e) and an increase in fall velocity, reaching $-2$ m s$^{-1}$ at the altitude of 1.3 km (Figs. 3a, 3c, 3d and 3f). Elevated values of DWR$_{Ka-W}$ presented in Figure 1b (more than 10 dB) below 3 km height before and after the RHI scan, indicate that particles are increasing in size in this layer. By combining all these information, we can provide an interpretation of this case study. From 5 km to 2.4 km height (Fig. 2b) VDPS identifies isometric particles, which are not correlated with any supercooled liquid droplets because only one spectral mode is detected in the spectrograms (Fig. 3). The temperatures in this altitude range increased from $-20\,°C$ at the top to $-8\,°C$ at its base, which points toward the occurrence of aggregation processes forming particles of spherical shape or low density. From 2.4 km to 2 km height, VDPS derives the presence of prolate particles which are transforming into isometric particles. Below 2 km height, three spectral modes can be identified in spectrograms (Figs. 3a and 3c) and one of them is associated with undetectable low values of SLDR. The correlation of this layer with a layer of low values of lidar VDR at 1.8 km height points toward the presence of supercooled liquid droplets. The other secondary spectral



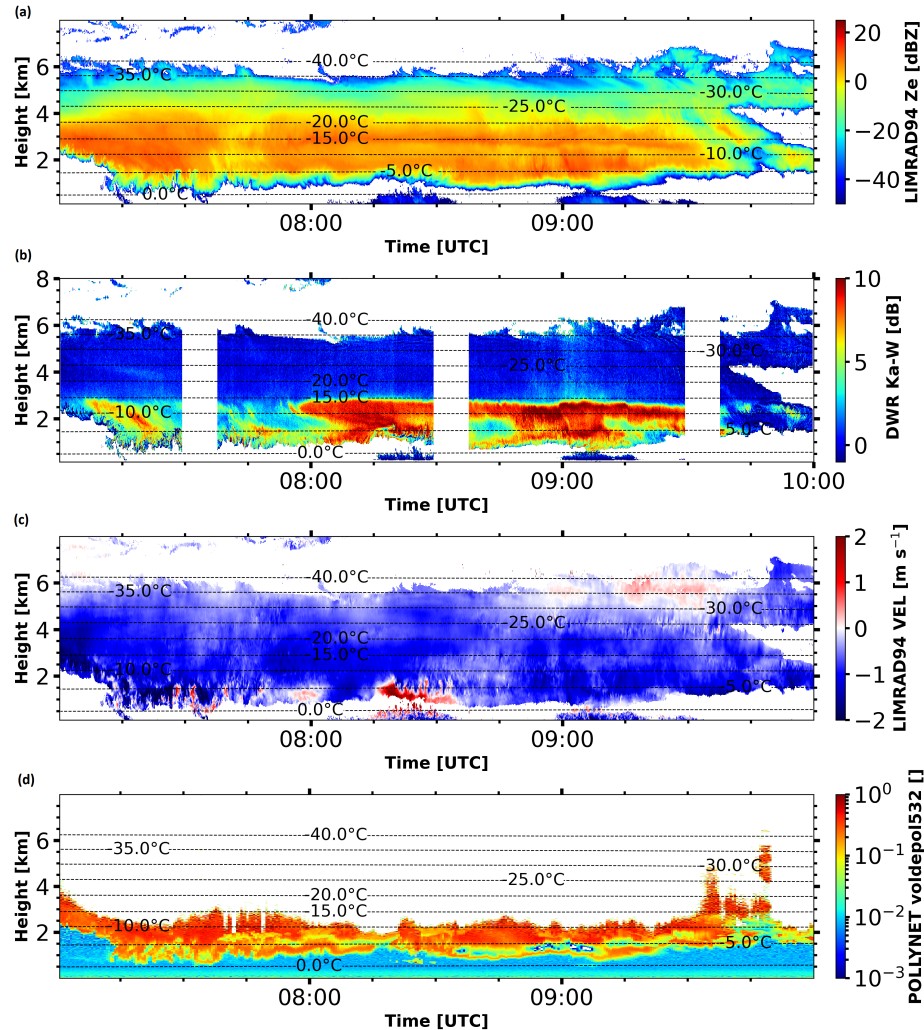

**Figure 4.** Similar to Fig. 1, but for a case study of a deep mixed-phase cloud event observed with multi-wavelength polarimetric cloud radars at Punta Arenas, Chile, on 30 August 2019 from 7:00 UTC to 10:00 UTC.

mode observable in the black rectangle in Figs. 3a and 3c, presents increased values of SLDR, indicating the presence of columnar ice crystals which are forming at temperatures between $-7\,°C$ and $-3\,°C$. In exactly this temperature range, Hallet-Mossop rime splintering is known to be most efficient in producing columnar crystals, which are falling slowly compared to the fast falling rimed main crystal population. Indeed, these rimed particles correspond to the main (fastest-falling) mode in the Doppler spectrograms, and to the isometric particles identified by VDPS. This is further confirmed by an increase in $Z_e$, $DWR_{Ka-W}$, and fall velocity below 2 km height, which indicates that particles of the main spectral mode are increasing in size and become more dense by falling.





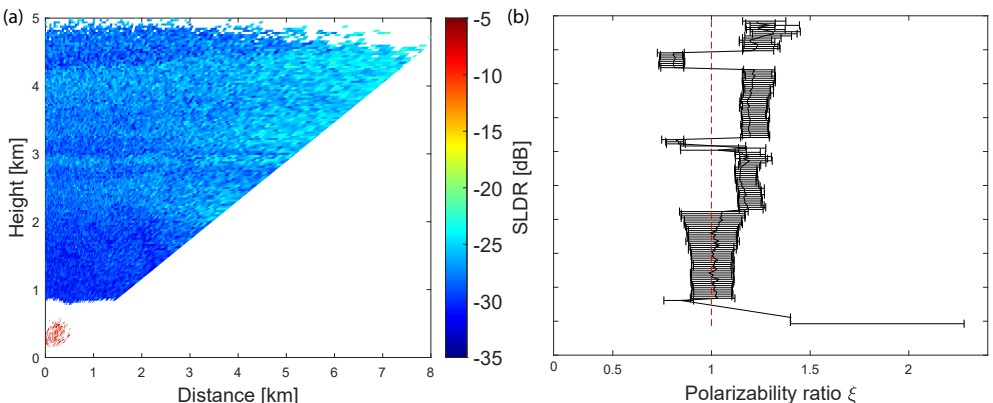

**Figure 5.** (a) RHI-scans of SLDR from $90°$ to $150°$ elevation angle observed on 30 August 2019, from 08:29:05 to 08:31:33 UTC in Punta Arenas, Chile. (b) Vertical distribution of the polarizability ratio $\xi$ derived from the VDPS method and correlated with the RHI scan presented in (a)

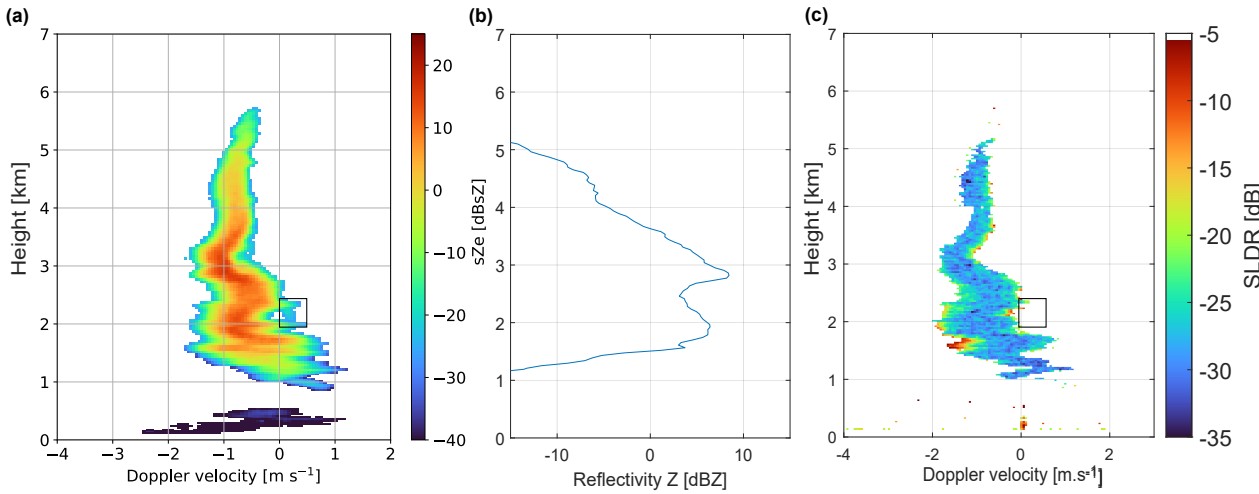

**Figure 6.** Spectrogram of (a) $Z_e$ recorded by LIMRAD94 at 08:29:14 UTC and (c) SLDR recorded by MIRA-35 at 08:29:09 UTC on 30 August 2019 in Punta Arenas, Chile. (b) represents the vertical distribution of $Z_e$ at 08:29:14 UTC. The black and red rectangles show the secondary spectral modes in the Doppler spectrograms



### 4.1.2 Riming case study on 30 August 2019, 08:30 UTC

Figure 4 shows a deep cloud system that passed the site between 7:00 UTC and 10:00 UTC. The white band at around 8:30 UTC in Fig. 4b represents the time period of the RHI scan of SLDR from $90°$ to $150°$ elevation angle that is shown in Fig. 5a. Figure 4c shows a strong variability of MDV, especially at cloud base. Thus, the fall velocity of particles cannot be used for interpretation of the case because of the turbulence.

In a first step, we will provide evidences for the presence of an aggregation layer that was present between 2.9 and 2.2 km height. At 2.9 km height, $Z_e$ abruptly decreases (Fig. 6b) which is associated with a co-located strong increase in $DWR_{Ka-W}$ of about 10 dB (Fig. 4b). This indicates that $Z_e$ at Ka-band as observed by MIRA-35 is not decreasing as strongly as $Z_e$ in W-band. Such a behavior is indicative of non-Rayleigh scattering, caused by large particles. Most likely, high $DWR_{Ka-W}$ highlights the formation of large aggregates which are known to form preferably at temperatures between $-10°C$ and $-20°C$ (Field and Heymsfield, 2003), as it was the case of the discussed layer. Above 2.2 km height, the polarizability ratio $\xi$ derived with the VDPS method (Fig. 5b) indicates the presence of slightly prolate quasi-isometric particles, only interrupted by two layers of slightly oblate hydrometeors at around 3 km and 4.5 km height ($\xi < 1$).

In a second step the measurement period will be screened for the potential presence of supercooled liquid droplets below the aggregate layer previously identified. As shown in Fig. 5a, SLDR above 2.2 km height is relatively high at all elevation angles (around $-25$ dB). At the height level of 2.2 km, SLDR suddenly drops to values below $-30$ dB. The associated microphysical change from non-spherical to isometric crystal shapes is also visible in Fig. 5b, where the polarizability ratio was found to be unity at all heights between 2.2 km and the cloud base ( $\xi \approx 1$), characterizing isometric particles. In reference to Fig. 6a, the isometric particles that were previously identified coincide with the presence of a secondary spectral mode at heights betweeen 2.2 and 2 km and Doppler velocities slightly above $0 \, \mathrm{m \, s^{-1}}$, evident in the $sZ_e$ Doppler spectrogram, as shown in the black box. This secondary spectral mode is not associated with any observable SLDR values (Fig. 6c), suggesting that liquid droplets are present. The extinction of the signal above a layer of low VDR measured by the lidar at 8:30 UTC (see Fig. 4d), corroborates the presence of a supercooled liquid layer at 2.2 km height. At the same time, $Z_e$ of the main hydrometeor population at zenith pointing increases from 2.2 km to 1.8 km height, reaching 8 dBZ, which indicates a growth in particle size (Fig. 6b). Below 1.8 km height, $Z_e$ decreases and the measured Doppler velocity increases due to turbulence at the base of the cloud, shown in Fig. 4c, or due to sublimation of ice particles. In Fig. 4b, the decrease in $DWR_{Ka-W}$ below 2.2 km height corresponds to the presence of the previously identified layer of supercooled liquid water at 2.2 km height, as detected by the lidar represented by a yellow layer before extinction (Fig. 4d), and LIMRAD94 in the black rectangle (Fig. 6a). This phenomenon indicates that large particles, such as large aggregates, are falling into a layer of supercooled liquid droplets and are forming smaller but more dense and isometric particles such as graupel. The most plausible scenario is that aggregates formed at around 2.9 km height, are falling through a layer of supercooled liquid droplets at around 2.2 km height, producing isometric graupel particles. This riming process occurs spontaneously at contact with the supercooled liquid droplets detected at 2.2 km height where particles will change quickly from prolate to isometric particles and where $DWR_{Ka-W}$ decreases suddenly at the same altitude. In this



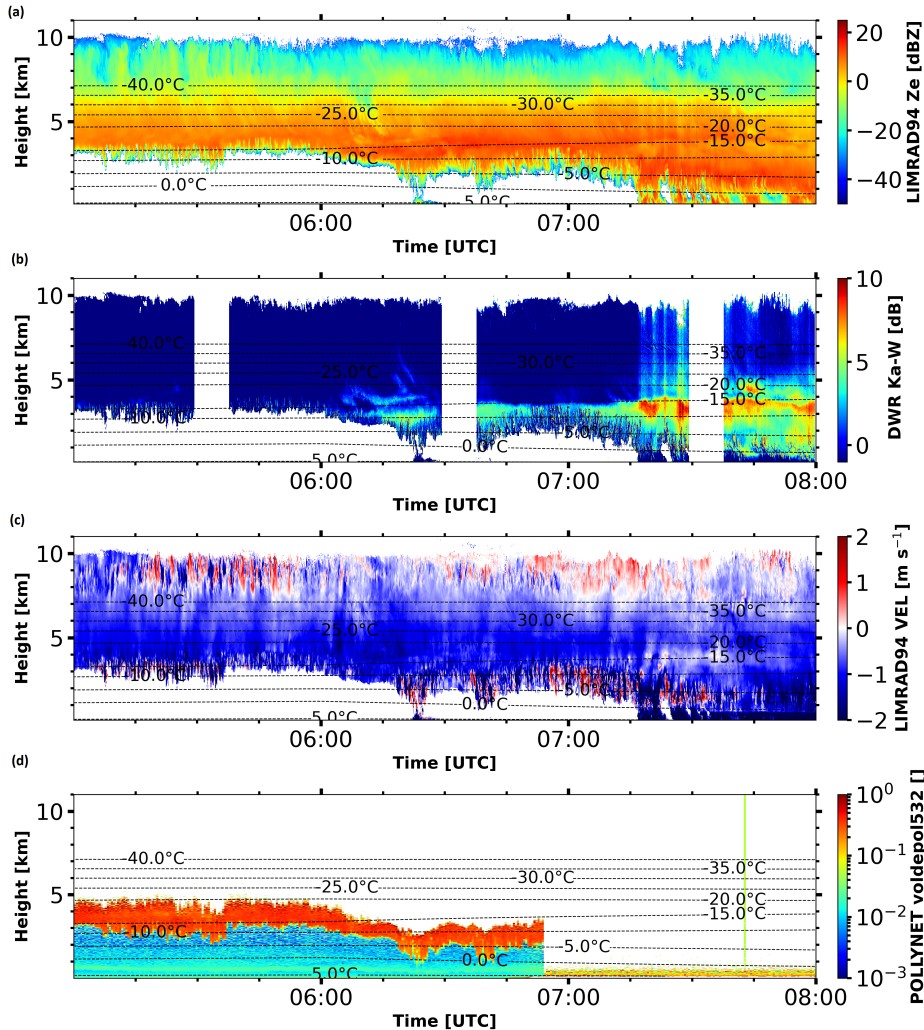

**Figure 7.** Similar to Fig. 1, but for a case study of a deep mixed-phase cloud event observed with multi-wavelength polarimetric cloud radars at Punta Arenas, Chile, on 26 August 2019 from 5:00 UTC to 8:00 UTC.

case, no indications for rime splintering are observed as the temperature (around $-10\,°\mathrm{C}$ at 2.2 km height, Fig. 4) is too low to allow the Hallett-Mossop process to occur.

**4.2 Aggregation detection**

The following section includes two case studies where aggregation was observed during DACAPO-PESO.




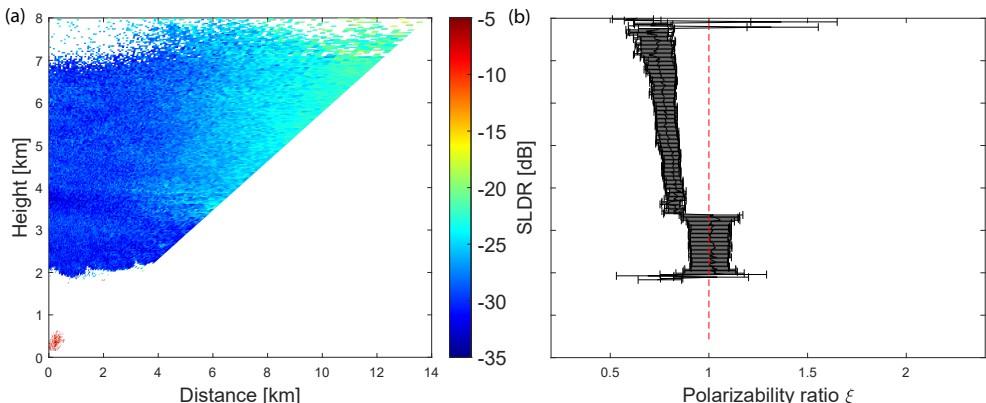

**Figure 8.** (a) RHI-scans of SLDR from $90°$ to $150°$ elevation angle observed on 26 August 2019, recorded from 06:29:05 to 06:31:33 UTC, in Punta Arenas, Chile. (b) Vertical distribution of the polarizability ratio $\xi$ derived with the VDPS method and correlated with the RHI scan presented in (a)

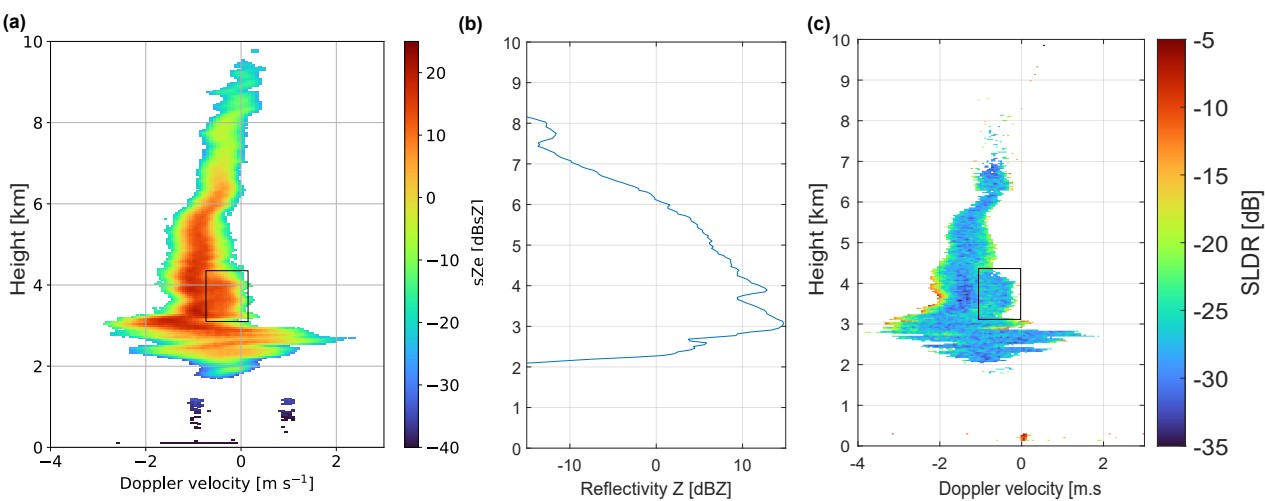

**Figure 9.** Spectrograms of (a) $Z_e$ recorded by LIMRAD94 at 06:29:23 UTC and (c) SLDR recorded by MIRA-35 at 06:29:24 on 26 August 2019 in Punta Arenas, Chile. (b) represents the vertical distribution of $Z_e$ at 06:29:23 UTC. The black rectangles shows the secondary spectral modes in the Doppler spectrograms





### 4.2.1 Aggregation case study on 26 August 2019, 06:30 UTC

A general overview is presented in Fig. 7 where the white band in the $DWR_{Ka-W}$ at 06:30 UTC (Fig. 7b) is due to the studied RHI scan. In Fig. 8a, the RHI scan shows an increase of SLDR from $90°$ to $150°$ elevation angle from the cloud top (at around

8 km height) to 3.2 km height. This increase is slightly more pronounced from 4 km to 3.2 km height associated with a polarizability ratio $\xi < 1$, hinting toward the presence of an oblate population of hydrometeors. Indeed, an additional slow-falling secondary mode is revealed in the cloud radar Doppler spectrogram of $sZ_e$ observed by LIMRAD94 at 06:29:23 UTC, shown in Fig. 9a. This secondary spectral mode can be identified in the same altitude region, marked by the black box, which confirms that two hydrometeor populations are coexisting. The secondary spectral mode depicted in Fig. 9a features elevated

values of $sZ_e$, of approximately 10 dBsZ, which is correlated with low values of SLDR, of around $-30$ dB, as shown in Fig. 9c, also marked by the black box. Low SLDR values at zenith pointing indicate that columnar crystals are unlikely (as SLDR would be higher) nor liquid droplets (as $sZ_e$ would be lower and SLDR not detectable). One hypothesis could be the formation of a second layer of oblate particles, which are falling slowly, such as dendritic crystals. In the discussed layer, the temperature ranges from $-20°C$ to $-10°C$, favouring the formation of planar (Libbrecht, 2003), dendritic crystals, which supports the

previous assumption.

At the base of the bi-modal cloud layer at 3.2 km height, a prominent shift of the polarizability ratio from $\xi = 0.7$ to $\xi \approx 1$ is observed, highlighting the presence of isometric particles below 3.2 km height. Indeed, in this layer below 3.2 km height, also SLDR is consistently low throughout the full RHI scan range from $90°$ to $150°$ elevation angle. From the vertical profile of Ze, shown in Fig. 9b, it can be seen that $Z_e$ reaches its maximum of 15 dBZ exactly at the height level of 3.2 km where the

shape transitioned from oblate to isometric. Concurrently, also $DWR_{Ka-W}$ of up to 5 dB (see Fig. 7b) is derived at and below this altitude. This indicates that a population of particles grows, increasing in size and fall velocity, while also becoming more isometric. Finally, the transition layer between oblate and isometric particles located at around 3.2 km height is not correlated with any supercooled liquid droplets, since the lidar observations do not show any liquid-related drops in the VDR (Fig. 7d). The absence of supercooled liquid droplets at 3.2 km height excludes the occurrence of riming processes as a cause of the

observed shift in the polarizability ratio below 3.2 km. To conclude, the observed trend from oblate to isometric particle shapes associated with an increase in $DWR_{Ka-W}$ and $Z_e$ without any signs of supercooled liquid droplets from LIMRAD94 or the lidar, strongly points to the occurrence of aggregation of ice particles during falling. Also, the temperature ranges from $-20°C$ and $-10°C$ (Fig. 7) and is typical for aggregation processes. To summarize, oblate particles like plates or dendrites collide and stick together by sedimentation which leads to the formation of aggregates. This process is accelerated by the passage of

these aggregates through the pronounced dendritic layer at around 4 km to 3.2 km height, favouring the production of larger aggregates, less dense particles appearing as isometric to the scanning SLDR-mode cloud radar.

### 4.2.2 Aggregation case study on 13 August 2019, 13:30 UTC

The studied cloud, depicted in Figure 10, is significantly influenced by orographic gravity waves, as is visible in the slowly alternating patterns of negative and positive Doppler velocity (Fig. 10c). Therefore, The MDV is not indicative of the particle

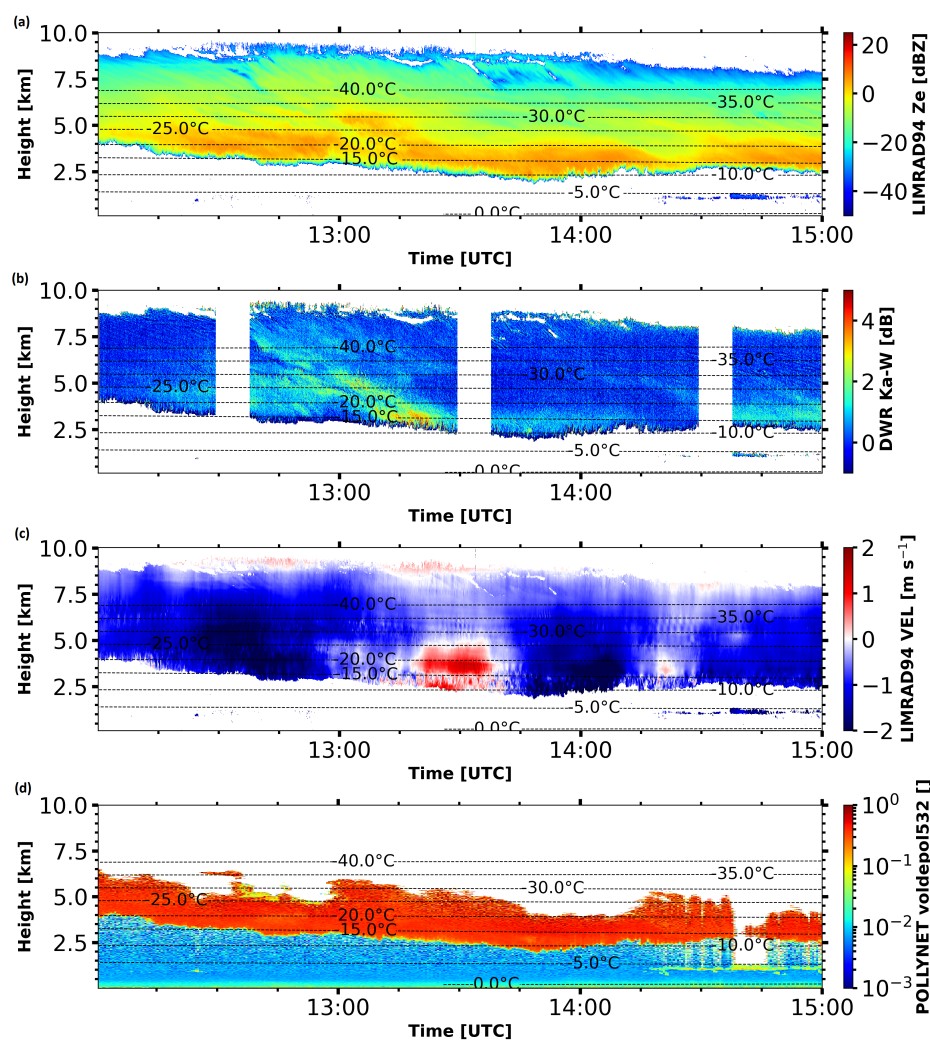

**Figure 10.** Similar to Fig. 1, but for a case study of a deep mixed-phase cloud event observed with multi-wavelength polarimetric cloud radars at Punta Arenas, Chile, on 13 August 2019 from 12:00 UTC to 15:00 UTC.



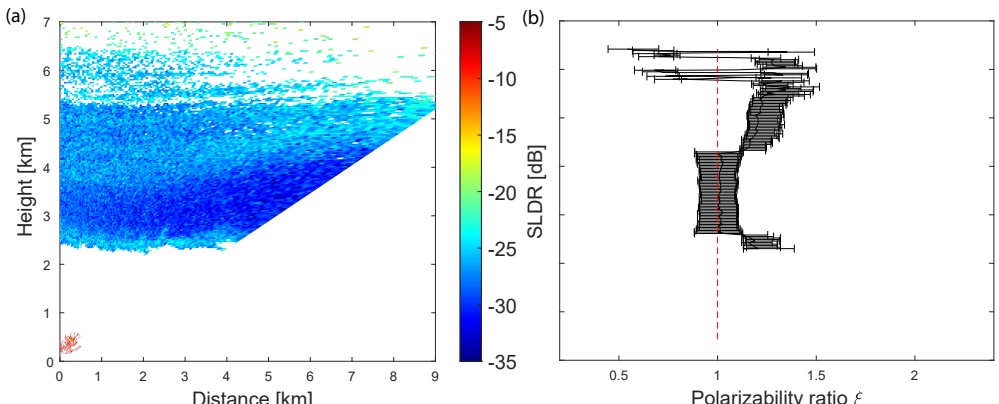

**Figure 11.** (a) RHI-scans of SLDR from $90°$ to $150°$ elevation angle observed on 13 August 2019, recorded from 13:29:05 to 13:31:33 UTC, in Punta Arenas, Chile. (b) Vertical distribution of the polarizability ratio $\xi$ derived with the VDPS method and correlated with the RHI scan presented in (a)

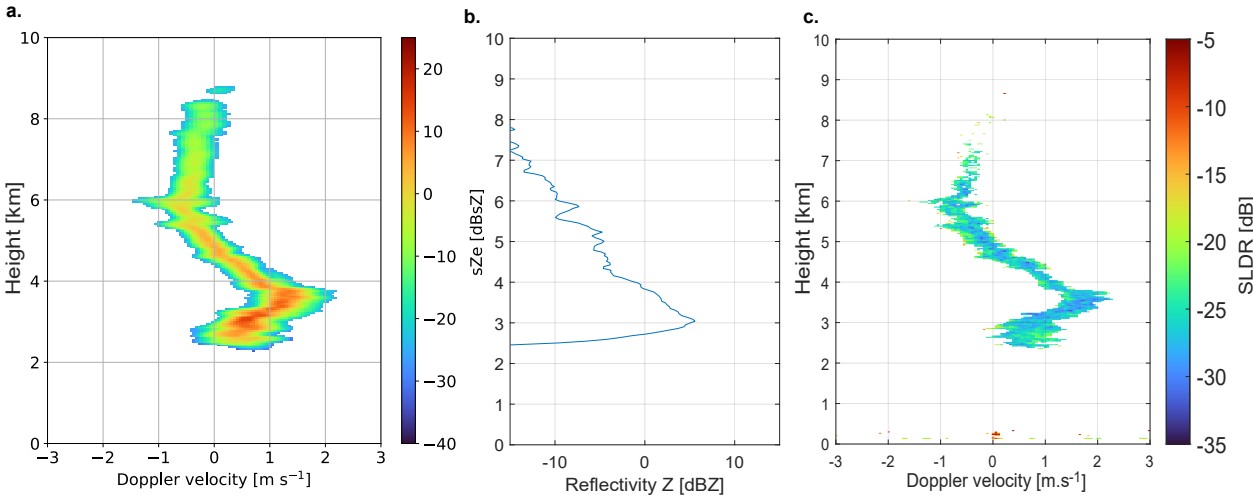

**Figure 12.** Spectrograms of (a) $Z_e$ recorded by LIMRAD94 at 13:29:39 UTC and (c) SLDR recorded by MIRA-35 at 06:29:24 on 13 August 2019 in Punta Arenas, Chile. (b) represents the vertical distribution of $Z_e$ at 13:29:39 UTC. The black rectangles shows the secondary spectral modes in the Doppler spectrograms



fall velocity, especially due to the updraft that occurred during the RHI scan at 13:30 UTC (second white band in Fig. 10b), and can thus not be used in this study. This case will thus provide another example about how even in the case of unavailability of MDV observations a combination of VDPS and spectral methods can be used to identify an aggregation event.

Figure 11a shows SLDR from $90°$ to $150°$ elevation angle during the studied RHI scan. In Figure 11a, two distinct layers are evident which are separated from each other at slightly above 4 km height. The layer below 4 km height is characterized

by low values of SLDR at all angles of the RHI scan while a certain elevation dependency in SLDR is observable above 4 km height. This limit layer presents a slightly curved structure of SLDR as a function of height which is visible along the range of elevation angles. This is likely a signature of an impact of the orographic gravity wave activity. While the gravity wave impact needs consideration within the study, it does not considerably affect the result delivered by the VDPS method presented in Figure 11b. Only the determination of the transition height between the two main layers within the presented case needs to

be handled with caution, as will be discussed below. Generally, the transition from the upper main layer to the lower one is well represented at approximately 4.2 km height in Figure 11b, where a transition from prolate to isometric particles, i.e., from $\xi > 1$ to $\xi \approx 1$, respectively, is derived with the VDPS method. However, it should be noted that the height of this transition layer has an uncertainty range of approximately 0.4 km, due to the elevation-dependent displacement of the SLDR signatures by the gravity waves. With respect to a potential presence of supercooled liquid water in the height range of the transition layer,

profiles of Doppler spectra and the lidar-derived VDR are evaluated next based on Figs. 12 and 10d, respectively. In the Doppler spectra observed by LIMRAD94, as illustrated in Figure 12a, no secondary spectral mode is visible. Also the lidar-measured VDR values are persistently elevated during the time frame from 13:00 to 14:00 UTC. From both, the spectral features and the VDR observation it can be concluded that no supercooled liquid water is present, at least up to the height of lidar signal attenuation at 5.5 km, as shown in Fig. 10d. This observation thus excludes the possibility of riming processes in this case.

Subsequent to the VDPS-based identification of a transition from prolate to isometric particles at around 4.2 km (Fig. 11b) and the absence of supercooled liquid water in this transition region, further indication for aggregation will be discussed as follows. In Figure 12b, an increase in $Z_e$ is noticeable from 4 km to 3 km height, correlated with low values of Mira-35-observed SLDR shown in Figure 12c. The $Z_e$ and SLDR signatures highlight that particles become larger in size and that their cross-section appears isometric at zenith pointing, which excludes the possibility that formation of columnar ice crystals by secondary ice

production occurred. This conclusion is corroborated by means of Figure 10b, where the measured $DWR_{Ka-W}$ below 4 km height increases by approximately 1.5 dB, pointing to the presence of small aggregates. The increase in particle size by the aggregation is correlated with the transition of particles toward isometric shapes at approximately 4.2 km height. As mentioned previously, the isometric aspect of particles can be due to a spherical shape or a low density of particles, which could well be the case for aggregates (associated in addition to high values of $Z_e$). To summarize, the polarizablity ratio values shown in

Figure 11b converge gradually from $\xi \approx 1.4$ to $\xi \approx 1$ from 5.5 km to 4.2 km height, indicating that prolate particles become isometric or appear isometric due to their low density. The transition from prolate to isometric particles is strongest at around 4.2 km height (the altitude has to be used with care because of the gravity waves mentioned before), where the presence of liquid water can be excluded based on spectograms detected by LIMRAD94 (Figure 12a), as well based on the absence of any low VDR values in the lidar observation (Fig. 10d). Below this layer, particles become larger as they fall (Figures 12b and



10b) pointing toward the occurrence of an aggregation process, producing particles with low density appearing as isometric particles to the radar. Considering the isolines of temperature shown in Fig. 10, it can be seen that the temperature of the identified isometric shape layer ranges from $-20\,^\circ$C to $-10\,^\circ$C, which is the preferred temperature range for the occurrence of aggregation processes.

## 5 Conclusions and outlook

In this study, the VDPS method (Teisseire et al., 2024) was combined with spectral retrieval techniques and polarization lidar observations to identify and investigate case studies of strong riming and aggregation. Earlier studies have demonstrated with success that polarimetric parameters, especially SLDR at different elevation angles, are useful to characterize the apparent shape of particles (Matrosov et al., 2001, 2012; Reinking et al., 2002; Myagkov et al., 2016a). The VDPS method is able to determine quantitatively the vertical distribution of particle shape but needs to be supplemented with other techniques to 410 differentiate riming and aggregation because both of these processes produce isometric particles: graupel, in the case of an advanced riming stage, are spherical and dense while aggregates are non-spherical particles with low density (low-refractive index) but also appear to be isometric. Detecting and associating supercooled liquid droplets with a transition layer where particles become isometric served as a method to identify riming processes. In addition, the $\mathrm{DWR_{Ka-W}}$ was used as a valuable tool for describing an increase in particle size (von Terzi et al., 2022). This study demonstrated by means of four case studies the 415 capability of the VDPS method to identify graupel and aggregates as isometric particles. In a subsequent step, these particles are differentiated between riming and aggregation processes using complementary tools such as spectrograms and $\mathrm{DWR_{Ka-W}}$.

Punta Arenas, where the investigated dataset was acquired, is a site where studies which deal with the discrimination of aggregation and riming are of utmost interest. At this site, orographic gravity waves are often observed. As a consequence, MDV cannot be used systematically to distinguish between riming and aggregation processes (Vogl et al., 2022). Therefore, 420 alternative tools need to be employed for this purpose. Second, the free troposphere of the southern hemisphere midlatitudes, where Punta Arenas is located, is prone to a high rate of occurrence of supercooled liquid water. It can be expected that the reported increased rates of occurrence of liquid water in cloud systems over this regions have a certain but yet unexplored impact on the cloud microphysical structure and precipitation formation. This study dealt with both of the motivating aspects. A new method was introduced to discriminate riming and aggregation even in the presence of atmospheric gravity waves, and 425 we provide first insights into the impacts of riming and aggregation in clouds over the investigated region.

Overall, two riming and two aggregation cases are discussed. The formation of graupel and aggregates shown in the four case studies is always associated with an isometric particle shape derived by the VDPS method. This finding demonstrates the relevance of the VDPS retrieval aiding in the detection of both graupel and aggregates, although it cannot distinguish unambiguously between the two when used as a stand-alone tool. The ECMWF-IFS air temperature range is an additional 430 tool to classify particles as graupel or aggregates. Indeed, riming processes occur preferentially at temperatures below $-10\,^\circ$C while aggregation processes are favored at temperatures between $-10\,^\circ$C and $-20\,^\circ$C. At first glance, the VDPS method can identify the presence of microphysical changes (e.g., riming, aggregation, sublimation), while spectral and/or multi-wavelength



techniques provide a more precise description of these changes. This is of great interest for research applications focusing on deepening the understanding of microphysical processes taking place in mixed-phase clouds. If applied to longer-term data
sets, this method could yield valuable insights about the processes driving the occurrence of riming vs. aggregation, e.g. by contrasting different sites with different climatic and aerosol properties.

Finally, the capability of the VDPS method to identify graupel and aggregates without differentiating them can simplify statistical work: initially, VDPS can sample all strong riming and aggregation cases, and subsequently, lidar or spectral techniques can differentiate them. This will drastically reduce the runtime of statistical analyses. It is thus goal of a follow-up study to pro-
vide a statistical evaluation of the occurrence of riming and aggregation over Punta Arenas based on the full DACAPO-PESO dataset and the methodology introduced herein.

*Code and data availability.*  The cloud-radar raw data, temperature data and lidar data are available upon request. Please contact the first or second author. Cloudnet data are available at https://cloudnet.fmi.fi. For plotting of the data, the tool pyLARDA was used, which is available at https://github.com/lacrostropos/larda. The VDPS algorithm is available upon request.

*Author contributions.*  AT developed the VDPS method, analysed the data and drafted the manuscript. All authors contributed to composing the manuscript.

*Competing interests.*  The contact author has declared that neither they nor their co-authors have any competing interests.

*Acknowledgements.*  Development of the VDPS method was funded by the Deutsche Forschungsgemeinschaft (DFG - German Research Foundation) project PICNICC (SE2464/1-1 and KA4162/2-1). The authors wish to thank the University of Magallan, Punta Arenas, Chile
for their logistic and infrastructural support during the LACROS deployment. We gratefully acknowledge the ACTRIS Cloud Remote Sensing Unit for making the Cloudnet datasets publicly available. We would also like to thank Martin Radenz and Andi Klamt for their assistance with PyLarda.



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
