# Peer review of "Attribution of riming and aggregation processes by application of the vertical distribution of particle shape (VDPS) and spectral retrieval techniques to cloud radar observations"

_EGUsphere, 2024_

## Author Comment (AC1)

**Response to the Reviewer #1**

Dear Editor,

We sincerely thank the Reviewer #1 for their insightful and constructive comments, which have greatly contributed to the improvement of our work. In particular, the comments regarding the figures were especially helpful in enhancing the reader's understanding of the Results section.

Some minor changes are visible in Fig. B, due to a recent improvement of the radar calibration concerning the antenna correction. This updated calibration aims to enhance data acquisition but does not affect the interpretation of case studies. The specific layers are more accurately defined due to the application of the new grid to Fig. B and the updated radar calibration.

The comments and suggestions provided by reviewer #1 and reviewer #2 have been thoroughly addressed in the updated manuscript. Consequently, we would like to submit the revised manuscript and the diff-version of the revised manuscript together with our responses to all the comments provided by reviewer #1 and reviewer #2. In our replies, all references to modified lines are given with respect to the diff-version of the manuscript.

Thank you for considering our work,

Best regards,

Audrey Teisseire, Anne-Claire Billault-Roux, Teresa Vogl and Patric Seifert.

Comments:

1. Each case study uses three figures illustrating: (Figure A) the time-series of zenith-pointing radar and lidar measurements, overlaid with temperature contours from the ECMWF analysis; (Figure B) RHI scans of SLDR and vertical distribution of the polarizability ratio from a short period within the case study, and (Figure C) Doppler spectra and profiles of reflectivity from a selected profile during the period relating to Figure B.

   The detailed discussion of the results requires frequent references to all of these figures, including specific height and temperature ranges, across the multiple figures, which each may have different ranges in the height axis.

   A few additional touches to the figures would greatly assist the reader in this cross-referencing and interpretation:
   o Please annotate on Figure A the time period of the RHI scans and profiles used in Figures B and C

   → We included dashed rectangles in Figs. A to highlight the time periods during which RHI scans and SLDR/Ze profiles were measured. We added this description to Fig. A: "*A dashed black rectangle in (a), (c) and (d) identifies the time period of RHI scans provided in Fig. B, and profiles used in Fig. C.*".

   o On Figures B and C, please include information about the critical temperature ranges which help us to interpret which processes are most likely: I leave it to

you to decide how to show this (e.g. duplicate y-axes, or an additional profile added to panel b) in both figures or an additional panel, or simply annotating selected height levels with the corresponding temperature)

→ We added a duplicate of the y-axis on the right side of the VDPS plot to display the corresponding temperature range.

- When specific layers are frequently referred to in the text, consider annotating these layers in Figures B and C

→ We included dashed black lines to Figures B and C which indicate the specific layers where the shift of the polarizability ratio toward isometric particles is observed.

- Please also consider adding subtitles to the panels of each figure to assist the reader when jumping back-and-forth between figures.

→ We added subtitles to all the figures presented in the Results section (section 4).

2. Some aspects of the interpretation of the case studies could be clarified and expanded upon:

1. In Section 4.1.2 the lowest layer is interpreted as "large aggregates…falling into a layer of supercooled liquid droplets and… forming smaller but more dense and isometric particles such as graupel". The possible mechanisms for a decrease in the size of the particles are not commented on. Are the low-density aggregates breaking up during the riming process? Does the turbulence near cloud-base indicate mixing with sub-saturated air leading to sublimation-driven breakup?

→ Thank you for the hint. In our opinion the more plausible scenario would be the transition of fluffy low-density aggregates to compact and high-density graupel. Indeed, the aggregated structure can be broken during the riming process, caused by impacts or subsequent freezing. The small parts of the initial porous aggregate collect efficiently supercooled liquid water drops, and are transformed into dense spherical graupel particles. In this case, the thermodynamic conditions presented for this case study are favorable for riming processes which is occurring fast and allow this transformation.

Turbulences observed around 8:15 UTC and 8:30 UTC visible in the variable vertical wind in Fig. 4c can favorize a riming event, promoting interactions between ice particle and supercooled liquid droplets. This interaction increases the collision frequency (which increases the likelihood of a transformation of porous aggregates to small high-density graupel, as described above). In Figure 4b, the decrease in Ze around 2.1 km height is initiated by the apparition of turbulences shown by the red to white colors related to MDV. In this case the turbulences can be responsible to the formation of small and high-density graupel derived by the VDPS method (Fig. 5b).

We added to lines 330-333: "*Indeed, this transformation is favoured by the presence of turbulence at cloud base (Fig. 4c), promoting interactions between ice particles and supercooled liquid droplets, and increasing the collision frequency. The small parts of the*

*initial porous aggregate collect efficiently supercooled liquid water drops, and are transformed into dense spherical graupel particles*."

2. In Section 4.2.1 the lowest layer, apparently containing spheroidal/low-density aggregates according to VDPS method (Figure 8b), is mostly discussed in terms of ruling out the presence of supercooled liquid and therefore of riming; however, the other observations characterising this layer include a very broad Doppler spectrum (Figure 9a) and indications of turbulence in the Doppler velocity (Figure 7c), as well as a rapid reduction of radar reflectivity (Figure 9b) toward cloud-base. Please comment on the interpretation of these features.

→ The broadening of the Doppler spectra is attributed to turbulence observed in the Mean Doppler Velocity (MDV) depicted in Fig. 7c in red. The decrease in Ze observed below 3 km height could be caused by the sublimation of aggregates making them smaller. Indeed, turbulence and shear zones are able to accelerate sublimation processes by increasing the exchange of heat and water vapor between ice surfaces and the environment. As discussed in Teisseire et al. (2024), the broadening of the Doppler spectra is negligible regarding polarimetric parameters such as SLDR (Matrosov et al, 2021). Therefore, VDPS is not impacted by this effect and derives isometric particles which sublimate quickly due to the strong turbulence located at the cloud base.

We propose to add this interpretation to lines 365-368: "*The decrease in Ze observed below 3 km height could be caused by the sublimation and associated shrinking of aggregates. Indeed, the Doppler spectrograms shown in Fig. 9a and Fig. 9c below 3 km height are broadened due to turbulence, potentially favoring the sublimation of ice particles. However, the broadening of Doppler spectrograms does not impact polarimetric parameters such as SLDR (Matrosov et al., 2021).*"

Typos and minor comments:

- L4: "harsh atmospheric conditions such as the presence of orographic gravity waves". Not sure if "harsh conditions" is the right word here; perhaps "adverse" with respect to Doppler-velocity based methods.

→ "*Harsh*" is changed to "*adverse*" in line 4.

- L5-6: "Core of the approach…" could be either "At the core of the approach…" or "Core to the approach…"

→ "Core of the approach" is changed in lines 6-7 to "*The core of the approach is to use profiles of the Vertical Distribution of Particle Shape (VDPS) method that serve as a proxy for identifying the presence of columnar, isometric, or prolate cloud particles*".

- L39: "low" instead of "small density"

→ "*small density*" is changed in line 39 to "*low density*".

- L46: "...by a lack of a efficient ice nucleating particles…" may have a missing word or a typo here.

→ The sentence is modified in line 46: "…*by a lack of efficient ice nucleating particles*…".

- L118: "...94-GHz radar and operates at 3.2-millimeter wavelength." This is already implicit in the 94-GHz frequency, so could read more like "...94-GHz radar (3.2-millimeter wavelength)."

→ Corrected as suggested in line 118.

- L269: "...by the presence low values of…" missing "of"

→ Corrected in line 275.

- L300: should be "...provide evidence of…"

→ Corrected in line 305.

---

## Author Comment (AC2)

**Response to the Reviewer #2**

Dear Editor,

We sincerely thank the Reviewer #2 for their insightful and constructive comments, which have greatly contributed to the improvement of our work. In particular, the comments regarding the figures were especially helpful in enhancing the reader's understanding of the Results section.

Some minor changes are visible in Fig. 2, 5 and 8, due to a recent improvement of the radar calibration concerning the antenna correction. This updated calibration aims to enhance data acquisition but does not affect the interpretation of case studies. The specific layers are more accurately defined due to the application of the new grid to Fig. 2, 5 and 8, and the updated radar calibration.

The comments and suggestions provided by reviewer #1 and reviewer #2 have been thoroughly addressed in the updated manuscript. Consequently, we would like to submit the revised manuscript and the diff-version of the revised manuscript together with our responses to all the comments provided by reviewer #1 and reviewer #2. In our replies, all references to modified lines are given with respect to the diff-version of the manuscript.

Thank you for considering our work,

Best regards,

Audrey Teisseire, Anne-Claire Billault-Roux, Teresa Vogl and Patric Seifert.

Section 3.2:

Do you have questions related to liquid water attenuation correction? How do you localize liquid layers in your measurements? Additionally, how do you distribute the column-integrated retrieval information of the LWP from the microwave radiometer over the columnar measurements of the radar? Do you also apply this to the scans of the Mira35? Since liquid water correction of radar measurements isn't a standard method, I would like to gain more insights into the process.

➔ The liquid water attenuation was not applied in this study. First, it is very challenging to perform a robust correction for liquid water attenuation. This process requires estimating a Liquid Water Content (LWC) profile, which is far from straightforward. Any errors in this estimation could significantly compromise the accuracy of the interpretations. Then, we underline that DWR is only used qualitatively in this study which implies that small errors caused by differential attenuation are acceptable. Moreover, only strong riming is studied in this article, making the use of DWR relevant without applying a correction for liquid water attenuation.
Last, the influence of liquid droplets is observed above the supercooled liquid layer, as indicated by vertical lines (visible in Fig. since 7:30 UTC). This study focuses on processes occurring below the supercooled liquid layers, where the presence of supercooled liquid droplets within the cloud does not interfere with the analysis.

Concerning RHI scans, we use a 35GHz radar (MIRA) and the attenuation is less pronounced compared to LIMRAD (94GHz radar). The liquid water attenuation is not applied to RHI scans and vertical profiles.

We propose to add this sentence to lines 193-194 : "*Since DWR is only used qualitatively in this study, no correction for liquid water attenuation was applied.*"

Section 4.

You mentioned that you use the LimRad94 Spectra to calculate the Ze values and other moments. Additionally, you utilize Doppler spectrograms for microphysical process analyses, as well as for detecting liquid layers and secondary ice production. Could you comment on how other noise clipping in the Doppler spectrum would influence your results? How did you set the noise threshold during processing? Providing such information would be helpful for the repeatability of these studies.

→ LIMRAD94 is an RPG radar, which can be set up to use spectral compression already during the measurement, i.e. only spectral lines exceeding a threshold T defined as

(1) $T = mean_{noise} + N \times std_{noise}$

are stored, with the mean and standard deviation of the noise derived using the technique described by Hildebrand and Sekhon (1974). The number of standard deviations N was set to 6 during LIMRAD94 data acquisition.

Regarding the question about the detectability of liquid and secondary ice peaks, we acknowledge that setting a noise threshold can definitely influence the results (e.g. missing small peaks which get removed by applying the threshold). We are however quite confident that we can detect most peaks quite reliably: The MIRA radar, which was employed right next to LIMRAD94, is a pulsed radar with higher sensitivity than LIMRAD94. In a recent study, Vogl and Radenz et al. (2024) compared liquid peaks detected in Doppler spectra recorded by these two different cloud radar systems in a case study from the same campaign and found a very high agreement between the two radar systems. (Fig. 4g and 4h in https://amt.copernicus.org/articles/17/6547/2024/)

We added to lines 124-128: "*LIMRAD94 can be set up to use spectral compression already during the measurement, i.e. only spectral lines exceeding a threshold T defined as*

$T = meannoise + N \times stdnoise$ *(1)*

*are stored, with the mean and standard deviation of the noise derived using the technique described by Hildebrand and Sekhon (1974). The number of standard deviations N was set to 6 during LIMRAD94 data acquisition.*"

Section 4.1.1.

Line 250: Can you briefly explain in the text what a white bad is?

→ We added to line 255: *", where no measurements are recorded at zenith pointing direction,"*

Section 4.1.2

Perhaps I missed it, but can you comment on why the Ze-time-height plot and the Ze e-profile of these cans show an increase in Ze from 4 to 3 km height? And why does there seem to be a minimum of Ze before the aggregation?

→ The reflectivity (Ze) begins to rise to about 0 dB at approximatively 3.5 km height, where the temperature is ranged at around -20°C, which suggests that the size or the number of particles increases in this region. It is challenging to precisely determine the specific process occurring within this layer and it is not in the scope of this study. Below this layer, at around 3 km height, particles aggregate slowly and form large aggregates (shown with high values of DWR in Fig. 4b).

Concerning the decrease in Ze from LIMRAD94, we have written in Lines 305-308: "*In a first step, we will provide evidence of the presence of an aggregation layer that was present between 2.9 and 2.2 km height. At 2.9 km height, Ze abruptly decreases (Fig. 6b) which is associated with a co-located strong increase in DWR Ka−W of about 10 dB (Fig. 4b). This indicates that Ze at Ka-band as observed by MIRA-35 is not decreasing as strongly as Ze in W-band. Such a behavior is indicative of non-Rayleigh scattering, caused by large particles.*". Examining Ze from MIRA reveals no minimum values around 2.6 km altitude, a discrepancy caused by non-Rayleigh scattering effects observed in LIMRAD94 measurements.

Section 4.2.1.

Line 339: I find it hard to identify the secondary mode in Fig. 9. Is the broadening of the spectra around 4 km really a second mode and not due to turbulence? Additionally, Figure 7c shows a large variation in the Doppler velocity field.

→ The secondary spectral mode is more pronounced in Fig. 9c within the SLDR Doppler spectra. The distinct double peak of low SLDR values is clearly observed between 4.2 and 3.5 km in altitude. The RHI scan at the same time presented in Fig. 1b below, processed with SLDR, calculated using the SNR in the cross-channel, highlights another population of ice crystals, specifically dendritic crystals (increase of SLDR from 90 to 150 degrees elevation angle). Indeed, if the main peaks are similar in the co- and cross-channels, it means that the main hydrometeor population depolarizes the most. On the other hand, the presence of different main peaks in the co- and cross-polarized Doppler spectra would imply the presence of a second hydrometeor population which depolarizes strongly, while still a non-polarizing hydrometeor population dominates the co-channel signal (Teisseire et al., 2024). This indicates that aggregates are the prominent ice crystal population and the dendritic layer is composed of the secondary ice crystal population which depolarizes the most compared to aggregates.

[Figure]

*Figure 1b: RHI scan of SLDR calculated using the main peak of SNR from the cross-channel, on 30 August 2019, from 08:29:05 to 08:31:33 UTC in Punta Arenas, Chile.*

Overall comments related to figures:

Is it possible to also get grids in the RHI scan figures and the polarizability ratio profiles? Identifying the corresponding heights is not always easy. As an alternative, one could indicate the regions of interest using circles or letters.

→ Thank you for the hint. We have inserted a grid to Figs. 2,5 and 8 to RHI and VDPS plots. Indeed, the specific layers are more accurately defined due to the application of the new grid.

Regarding the SLDR figures, the contrast of the plots is not the largest. Is it possible to use a color bar where the signatures you want to show are more prominent?

→ We adjusted the colorbar limits in Figs. 2a,5a and 8a to enhance contrast and make the signatures more visually distinct.